# Impact of Placental SLC2A3 Deficiency during the First-Half of Gestation

**DOI:** 10.3390/ijms232012530

**Published:** 2022-10-19

**Authors:** Cameron S. Lynch, Victoria C. Kennedy, Amelia R. Tanner, Asghar Ali, Quinton A. Winger, Paul J. Rozance, Russell V. Anthony

**Affiliations:** 1College of Veterinary Medicine, Colorado State University, Fort Collins, CO 80523, USA; 2Anschutz Medical Campus, University of Colorado, Aurora, CO 80045, USA

**Keywords:** placenta, glucose uptake, SLC2A3, SLC2A1, insulin, glucagon

## Abstract

In the ruminant placenta, glucose uptake and transfer are mediated by facilitative glucose transporters SLC2A1 (GLUT1) and SLC2A3 (GLUT3). SLC2A1 is located on the basolateral trophoblast membrane, whereas SLC2A3 is located solely on the maternal-facing, apical trophoblast membrane. While SLC2A3 is less abundant than SLC2A1, SLC2A3 has a five-fold greater affinity and transport capacity. Based on its location, SLC2A3 likely plays a significant role in the uptake of glucose into the trophoblast. Fetal hypoglycemia is a hallmark of fetal growth restriction (FGR), and as such, any deficiency in SLC2A3 could impact trophoblast glucose uptake and transfer to the fetus, thus potentially setting the stage for FGR. By utilizing in vivo placenta-specific lentiviral-mediated RNA interference (RNAi) in sheep, we were able to significantly diminish (*p* ≤ 0.05) placental SLC2A3 concentration, and determine the impact at mid-gestation (75 dGA). In response to SLC2A3 RNAi (*n* = 6), the fetuses were hypoglycemic (*p* ≤ 0.05), exhibited reduced fetal growth, including reduced fetal pancreas weight (*p* ≤ 0.05), which was associated with reduced umbilical artery insulin and glucagon concentrations, when compared to the non-targeting sequence (NTS) RNAi controls (*n* = 6). By contrast, fetal liver weights were not impacted, nor were umbilical artery concentrations of IGF1, possibly resulting from a 70% increase (*p* ≤ 0.05) in umbilical vein chorionic somatomammotropin (CSH) concentrations. Thus, during the first half of gestation, a deficiency in SLC2A3 results in fetal hypoglycemia, reduced fetal development, and altered metabolic hormone concentrations. These results suggest that SLC2A3 may be the rate-limiting placental glucose transporter during the first-half of gestation in sheep.

## 1. Introduction

Glucose is the primary energy substrate for fetal oxidative processes and growth [1]. Due to a lack of endogenous fetal glucose production, until near term, the maternal circulation is the only source of glucose for the placenta and fetus [2,3]. Placental glucose uptake and transfer to the fetus requires a positive maternal-to-fetal glucose concentration gradient that is mediated by facilitative glucose transporter (GLUT) proteins on both the maternal-facing apical microvillus and fetal-facing basal trophoblast membranes. In sheep, SLC2A3 (GLUT3) is localized to the apical microvillus trophoblast membrane, while SLC2A1 (GLUT1) is localized to the basolateral trophoblast membrane, and as such, both transporters must be utilized sequentially for glucose uptake and transfer from maternal to fetal circulation [4]. While both SLC2A1 and SLC2A3 are present throughout gestation [5], SLC2A1 has been regarded as the primary placental glucose transporter as it is the most abundant glucose transporter in the mammalian placenta and increases in abundance as gestation progresses [6]. Conversely, SLC2A3 is less abundant than SLC2A1, but SLC2A3 has a five-fold greater affinity and transport capacity for glucose [7].

Functional placental insufficiency is a major cause of fetal growth restriction (FGR), however, the specific causes of placental insufficiency are not well characterized. A common hallmark of FGR is reduced placental transfer of glucose, resulting in the fetuses becoming hypoglycemic [8,9]. When assessed in pre-term or term FGR placentas, there is a lack of down-regulation of either SLC2A1 or SLC2A3 [10,11], indicating that the fetal hypoglycemia does not stem from a deficit in glucose transport mechanisms. However, these observations may not reflect the glucose transport capacity of the FGR placenta throughout gestation. Any deficit in placental glucose transport during the first-half of pregnancy could impact placental development and function, thus potentially resulting in functional placental insufficiency and setting the stage for fetal hypoglycemia and FGR. For obvious ethical reasons, the relative importance of SLC2A1 and SLC2A3 in placental glucose transport at different stages of gestation, and how a deficit in either glucose transporter may alter glucose transport, cannot be addressed in humans. As SLC2A1 is localized solely to the basolateral trophoblast membrane and SLC2A3 is localized solely to the maternal-facing apical microvillous membrane [4], the sheep placenta provides the opportunity to differentiate the relative importance of each placental glucose transporter as well as apical versus basolateral transport. 

The development of in vivo lentiviral-mediated RNA interference (RNAi) [12,13,14,15,16,17], that specifically targets trophoblast cells, provides the opportunity to directly assess the relative importance of placental SLC2A3. While the abundance of SLC2A3 is less than SLC2A1 [5], SLC2A3 appears to play an important role in trophoblast uptake of glucose due to its location on the maternal-facing apical trophoblast membrane [4] and its five-fold greater affinity and transport capacity for glucose [7]. Thus, any deficit in SLC2A3 may have a major impact on placental uptake of glucose, placental development and function, and fetal development. Accordingly, we hypothesized that SLC2A3 deficiency would result in impaired placental development and significant FGR by mid-gestation (75 dGA). Therefore, it was our objective to use lentiviral-mediated RNAi to attenuate the expression of placental SLC2A3 to assess its relative importance in placental glucose transport, as assessed at mid-gestation.

## 2. Results

### 2.1. RNA Interference of SLC2A3 in iOTR Cells

To assess the effectiveness of the SLC2A3 RNAi construct, iOTR cells were infected at a MOI of 500 with either the NTS RNAi or SLC2A3 RNAi lentivirus, approximately the MOI used for blastocyst infection, and SLC2A3 concentrations were determined. SLC2A3 RNAi resulted in a 91% reduction in SLC2A3 concentration (*p* ≤ 0.05; Figure 1) as compared to NTS RNAi-infected iOTR cells. 

### 2.2. Fetal and Placental Measurements at Mid-Gestation

At 70 dGA, as assessed by ultrasonography, fetal binocular distance tended to be reduced (*p* ≤ 0.10), and both femur length and tibia length were significantly shorter (*p* ≤ 0.05), whereas there appeared to be no impact of SLC2A3 RNAi on fetal crown-rump length or abdominal circumference (Table 1). Doppler assessment of umbilical artery velocimetry revealed no differences in umbilical artery pulsatility indices, resistance indices, systolic/diastolic ratios, fetal heart rates, umbilical artery cross-sectional areas, or cross-sectional diameters between NTS RNAi and SLC2A3 RNAi fetuses (*p* ≥ 0.10; Table 1). When assessed at the 75 dGA terminal surgery, head circumference, femur length, and tibia length were all significantly reduced in the SLC2A3 RNAi pregnancies (*p* ≤ 0.05; Table 2), and fetal weight tended (*p* ≤ 0.10) to be less (Table 2), whereas crown-rump length and abdominal circumference were not impacted by SLC2A3 RNAi. While fetal liver weight was not impacted (Table 2), fetal pancreas weight was significantly reduced (*p* ≤ 0.05). Placental weight (total placentome weight) was less, but did not reach statistical significance (*p* ≥ 0.10), and placentome number was not impacted by SLC2A3 RNAi (Table 2).

SLC2A3 RNAi resulted in a 37% reduction in placental SLC2A3 (*p* ≤ 0.05; Figure 2), as well as a 38% increase in SLC2A1 concentration (*p* ≤ 0.10; Figure 2) in the SLC2A3 RNAi pregnancies. Additionally, there was no effect of SLC2A3 RNAi on placental SLC2A8 (*p* ≥ 0.10; Figure 2). 

### 2.3. Maternal and Fetal Plasma Measurements at Mid-Gestation

At 75 dGA, uterine artery and vein concentrations of glucose (Figure 3) and lactate (Figure 3) were not significantly impacted by SLC2A3 RNAi (*p* ≥ 0.10), although SLC2A3 RNAi maternal glucose concentrations were 20–25% lower. In contrast, umbilical vein and artery glucose concentrations were significantly reduced by 42% and 46%, respectively, in the SLC2A3 RNAi pregnancies (*p* ≤ 0.05; Figure 4). There were no statistical differences observed in lactate concentrations in either the umbilical vein or artery (*p* ≥ 0.10; Figure 4). Individual amino acid concentrations in the uterine and umbilical vasculature are presented in Table 3 and Table 4. There were few SLC2A3 RNAi-induced changes in maternal plasma amino acid concentrations, with the exceptions being greater (*p* ≤ 0.05) concentrations of asparagine and lysine in the uterine artery, and tendencies (*p* ≤ 0.10) for increased valine and ornithine in the uterine artery and citrulline in the uterine vein. Similarly, there were few changes in fetal plasma amino acids, other than significant reductions (*p* ≤ 0.05) in the umbilical vein and artery concentrations of arginine, and increased (*p* ≤ 0.05) asparagine in the umbilical artery.

As evidenced in Figure 5, SLC2A3 RNAi did not impact uterine artery insulin or uterine vein CSH concentrations, but did result in a significant reduction (*p* ≤ 0.05) in the uterine artery concentrations of both glucagon and IGF1. In contrast, both umbilical artery insulin and glucagon (Figure 6) were reduced (*p* ≤ 0.10) 44% and 53%, respectively, whereas umbilical artery IGF1 was not impacted by SLC2A3 RNAi, nor was fetal liver INSR concentration (data not presented). However, umbilical vein concentrations of CSH were increased 70% (*p* ≤ 0.05; Figure 6) in SLC2A3 RNAi pregnancies.

### 2.4. Placental mRNA Concentration of the Insulin-like Growth Factor Axis

Placental tissues harvested at 75 dGA were assessed for IGF, IGFBP, and IGFR (IGF receptor) mRNA concentrations. While there were no differences in placental IGF1 mRNA concentration (*p* ≥ 0.10; Table 5), placental IGF2 mRNA concentration was increased by 71% (*p* ≤ 0.05; Figure 7) in SLC2A3 RNAi pregnancies. Additionally, in SLC2A3 RNAi pregnancies, placental IGF1R and IGF2R mRNA concentrations were increased by 40% and 69% (*p* ≤ 0.05; Figure 7), respectively. There were no differences in placental IGFBP1, IGFBP2, or IGFBP3 mRNA concentrations between treatments (*p* ≥ 0.10; Table 5).

## 3. Discussion

As glucose is the primary energy substrate supporting fetal development [1], the fetus is reliant upon placental glucose uptake and transfer, mediated by facilitative glucose transporters. The importance of placental glucose transfer is exemplified in FGR pregnancies, with the magnitude of fetal hypoglycemia being correlated with the severity of FGR [8,9]. SLC2A1 (GLUT1) and SLC2A3 (GLUT3) are the primary transporters in human and ruminant placenta [4,18] believed responsible for uptake and transfer, yet the relative importance of each is debatable. In humans, SLC2A1 is found in both the microvillous (apical) and basal membranes of the syncytiotrophoblast [6,19], whereas SLC2A3 is localized to just the microvillous membrane [20]. By contrast, in sheep, SLC2A1 is localized to the basolateral trophoblast membrane and SLC2A3 is localized to the microvillous trophoblast membrane [4]. Beyond the individual transporters, the more important question is whether microvillous glucose transport is more or less important than basal membrane transport. In vivo maternal and fetal glucose clamp studies led to the conclusion that placental glucose transport capacity is greater on the fetal surface than the maternal surface [21]. However, the placenta is a highly metabolic organ in itself, and placental glucose utilization accounted for 80 and 72% of uterine glucose uptake at mid- and late-gestation, respectively [22,23], directly impacting the maternal-fetal glucose gradient, thereby requiring sufficient microvillous glucose uptake to maintain placental function.

As SLC2A3 is localized on the apical microvillous membrane in both sheep and humans [4,20], SLC2A3 likely plays an important role in trophoblast uptake of glucose in both species. Accordingly, we used our lentiviral-mediated in vivo RNAi methods [12,13,14,15,16,17] to diminish SLC2A3 in the sheep placenta to evaluate the relative importance of microvillous trophoblast glucose uptake during the first half of gestation. SLC2A3 RNAi resulted in a 37% reduction (Figure 2) in placental SLC2A3 concentration at 75 dGA. This reduction in SLC2A3 was sufficient to induce significant fetal hypoglycemia (Figure 4) and reduce fetal growth (Table 4), as assessed at mid-gestation. In contrast, SLC2A1 was increased (Figure 2), which we hypothesize is in an attempt to offset the deficit in SLC2A3 in terms of glucose uptake and transport. In mice, the homozygous *Slc2a3^−/−^* genotype results in embryonic lethality, whereas the heterozygous *Slc2a3^+/−^* genotype resulted in late-gestation FGR [24], further supporting the requirement of SLC2A3. In humans, placental SLC2A3 is more abundant in early gestation [20], and it has been suggested that its greater affinity and glucose transport capacity may be important during the early stages of gestation when glucose delivery to the developing placenta is low [20]. Interestingly, during late-gestation in FGR pregnancies, an upregulation of SLC2A3 is thought to be an adaptive response to increase placental glucose uptake and transfer [11,25,26]. Our results further support the important role of placental SLC2A3, and that a deficiency in microvillous trophoblast glucose uptake is crucial in supplying the fetus adequate glucose during the first half of gestation.

As noted above, SLC2A3 RNAi resulted in significant fetal hypoglycemia. Due to these pregnancies being studied at mid-gestation, they did not undergo maternal and fetal catheterization that would have allowed steady-state assessment of uterine and umbilical blood flows, as well as uterine and umbilical uptakes and uteroplacental utilization of nutrients [27]. A common hallmark of FGR pregnancies is fetal hypoxia and increased fetal lactate concentrations [28,29]. While we were unable to quantify oxygen content in the collected blood samples, there was a 48% increase in umbilical artery lactate concentrations in SLC2A3 pregnancies, although this was not statistically different (*p* = 0.14). As the majority of fetal lactate is produced by the fetus [30,31], as evidenced by greater lactate concentrations in the umbilical artery as compared to the umbilical vein (Figure 4), these data indicate that the increase in umbilical artery lactate is the result of fetal hypoxia. However, as uterine and umbilical blood flows are the major determinants of fetal oxygen delivery [32], the lack of differences observed in the 70 dGA Doppler velocimetry assessment would suggest that umbilical blood flow in the SLC2A3 RNAi pregnancies, is not restricting fetal oxygen delivery. We did assess amino acid concentrations in the uterine and umbilical blood samples collected and there did not appear to be an overall impact of SLC2A3 RNAi on amino acid concentrations. This would indicate that there was not an increase in amino acid oxidation in response to fetal hypoglycemia, to maintain fetal oxidative metabolism [33,34]. Using maternal hyperinsulinemia clamps to induce fetal hypoglycemia, DiGiacomo and Hay [35], demonstrated that fetal oxygen consumption was reduced proportionally to fetal hypoglycemia and the rate of fetal growth reduction, diminishing the likelihood of increased amino acid oxidation. This was further demonstrated, following 8 weeks of fetal hypoglycemia, in which it was determined that fetal plasma leucine disposal, leucine flux into protein synthesis, and leucine oxidation were not impacted by fetal hypoglycemia [36].

There were impacts of SLC2A3 RNAi on fetal growth, which may have resulted solely from the hypoglycemia. Notably, fetal pancreas weight was reduced 23% (*p* ≤ 0.05). As expected, umbilical artery insulin was reduced 44% (*p* ≤ 0.10), essentially equivalent to the reduction in umbilical glucose concentrations. Surprisingly, umbilical artery glucagon concentrations were also reduced (*p* ≤ 0.10) in the SLC2A3 RNAi pregnancies. The fact that both insulin and glucagon concentrations were reduced suggests that the fetal hypoglycemia induced by SLC2A3 RNAi during the first half of gestation had an overall effect on pancreas development and growth, rather than a β cell-specific effect. Glucose-stimulated insulin secretion at mid-gestation is 20% of the rate near term [37], indicating that during the first half of gestation the pancreas may not be responding to glucose concentrations in the specific fashion that occurs during the second half of gestation. Notably, arginine-induced insulin secretion increases with gestational age in a similar fashion to glucose-stimulated insulin secretion [37], and of all of the amino acids assessed, only arginine was significantly reduced (*p* ≤ 0.05) in both the umbilical vein and artery.

In various experimental models that produce functional placental insufficiency that results in FGR, fetal hypoglycemia, fetal hypoinsulinemia, and decreased fetal liver growth are common characteristics [13,15,28,38,39]. Additionally, when fetal liver growth is decreased in FGR pregnancies, a decrease in IGF1 concentrations is often observed [13,15,40], as well as an upregulation of the fetal liver INSR in response to fetal hypoinsulinemia [41]. However, in response to SLC2A3 RNAi, fetal liver weight, INSR concentration, and umbilical artery IGF1 concentrations were unaffected. The lack of impact on the fetal liver may be due in part to the 70% increase in umbilical vein CSH (Figure 6) observed in the SLC2A3 RNAi pregnancies. A similar increase in umbilical CSH, in response to fetal hypoglycemia and hypoinsulinemia, was reported in fasted late-gestation pregnant ewes that was then subsequently reversed upon refeeding of the ewes [42]. The increase in circulating CSH concentrations may have preserved fetal liver weight and function, as CSH deficiency results in reduced fetal liver weights during early [17] and late-gestation sheep pregnancies [13,14], as well as significant reductions in umbilical artery IGF1 concentrations [13,14]. Collectively, these data may suggest that by enhancing umbilical concentrations of CSH, the placenta indirectly salvages fetal liver growth and function in the face of fetal hypoglycemia, at least during the first half of gestation.

With the SLC2A3 pregnancies, placental weight was reduced 21%, which did not reach statistical significance (*p* = 0.13). Within the placenta, *IGF2*, *IGF1R* and *IGF2R* mRNA concentrations were the only members of the insulin-like growth factor axis impacted, and all three mRNA were significantly (*p* ≤ 0.05) elevated in SLC2A3 RNAi placenta (Figure 7). The upregulation of these mRNA may be a compensatory mechanism to stimulate placental growth, thus increasing the total nutrient exchange surface area to combat fetal hypoglycemia. Targeted mutagenesis studies have demonstrated that placental IGF2 plays a role in modulating placental growth as overexpression of *Igf2* results in placental overgrowth [43] and total ablation of *Igf2* results in placental growth restriction [44]. Placental IGF2 mediates its effects through IGF1R, as IGF2R has been demonstrated to be a clearance receptor for IGF2 [45,46] that prevents IGF2 from overstimulating IGF1R and producing placental overgrowth [46,47]. A similar increase in placental *IGF2* expression has been shown to be an adaptive response to decreased placental growth in other sheep FGR models [48,49].

Placenta-specific SLC2A3 RNAi did not have a statistically significant impact on uterine artery or vein glucose and lactate concentrations (Figure 3), but numerically there was a 20–25% reduction in uterine blood glucose concentrations in SLC2A3 RNAi pregnancies. In contrast to the significant changes observed in umbilical circulation, uterine artery insulin and uterine vein CSH concentrations were not impacted by SLC2A3 RNAi (Figure 5). However, to our surprise, the uterine artery concentrations of both IGF1 and glucagon were significantly (*p* ≤ 0.05) reduced in SLC2A3 RNAi pregnancies. We did not hypothesize that there would be an impact on maternal hormone concentrations as the use of replication-deficient lentivirus to infect hatched blastocysts results in the RNAi being limited to the trophectoderm lineage of the placenta [12,50,51], such that the RNAi is placenta-specific. The non-significant reductions in maternal glucose may be tied to the reduction in maternal glucagon, but what is driving the reduction in maternal IGF1 and glucagon is not apparent. We can only speculate that this resulted from reductions in trophoblast uptake of glucose, altering placental secretory products that impact maternal IGF1 and glucagon.

Since SLC2A3 is limited to the microvillous membrane of the placental trophoblast in sheep [4], using lentiviral-mediated RNAi, we were able to assess the impact of SLC2A3 deficiency during the first half of gestation. Our results confirm that microvillous glucose uptake can be rate-limiting to fetal growth and development during early gestation, while not fully resolving whether microvillous or basal glucose transport is more important. Some of our results could be viewed as being predictable, in response to limiting glucose transfer to the fetus, while others were not. The “global” impact of fetal hypoglycemia on pancreas growth and function, with both insulin and glucagon concentrations being diminished, were not expected, especially since fetal liver growth and IGF1 secretion were not affected. The most surprising result was the diminished uterine artery IGF1 and glucagon concentrations. While considerable effort has been expended on examining maternal glucose, insulin and IGF1 in normal and compromised pregnancies, the role of maternal glucagon for the most part has been overlooked, although Qiao et al. [52] recently reported that pregnancy in mice induces an expansion of maternal α-cell mass, and an increase in maternal glucagon concentrations during early pregnancy. Furthermore, the demonstration [53] that fetal hyperglucagonemia during late-gestation results in significant reductions in uterine artery blood flow and placental CSH production and secretion into maternal circulation, without impacting umbilical blood flow or CSH concentrations, highlights not only the importance of glucagon during pregnancy, but also that the three compartments (maternal, placental and fetal) are intimately integrated and need to be investigated together [54]. This research, therefore, also exemplifies the utility of integrating in vivo RNAi in an animal model that can allow steady-state assessment of altered maternal-placental-fetal physiology [27].

## 4. Materials and Methods

All procedures conducted with animals were approved by the Colorado State University Institutional Animal Care and Use Committee (Protocol 1483), as well as the Institutional Biosafety Committee (17-039B).

### 4.1. Lentiviral Generation

Lentiviral infection was used to stably integrate and express shRNA targeting *SLC2A3* mRNA in the host cell. The shRNA sequences for hLL3.7 472 (SLC2A3 RNAi) and hLL3.7 NTS (non-targeting sequence; control RNAi) constructs are presented in Table 6. All subsequent virus generation and titering followed the procedures extensively described previously [13].

### 4.2. Generation of SLC2A3 RNAi Pregnancies

All ewes (Dorper breed composition) were group housed in pens at the Colorado State University Animal Reproduction and Biotechnology Laboratory, and were provided access to hay, trace minerals, and water to meet or slightly exceed their National Research Council [55] requirements. Animal management, estrus synchronization, and embryo transfers were done as previously described [13,14,15]. In summary, after synchronization and subsequent breeding, at 9 days post-conception, donor ewes were euthanized (88 mg/kg Euthasol; VetOne, Conshohocken, PA) and the uteri were harvested and flushed to collect hatched and fully expanded blastocysts. Each blastocyst was infected with 150,000 transducing units of either NTS RNAi or SLC2A3 RNAi virus as previously described [13,14,15]. Following 5 h of incubation with the virus, each blastocyst was washed and a single blastocyst was surgically transferred into the uterine horn ipsilateral to the corpus luteum of a synchronized recipient ewe. All recipient ewes (NTS RNAi *n* = 10; SLC2A3 RNAi *n* = 13) were then monitored daily for return to standing estrus and confirmed pregnant at 50 days of gestational age (dGA) by ultrasonography (Mindray Medical Equipment, Mahway, NJ, USA). At 70 dGA, all successful pregnancies (6 NTS RNAi and 6 SLC2A3 RNAi) underwent Doppler velocimetry assessment as described previously [14].

### 4.3. Tissue Collection

At 75 dGA, six NTS RNAi (5 males and 1 female) and six SLC2A3 RNAi (3 males and 3 females) pregnancies underwent a terminal surgery as previously described [13]. In summary, pregnant recipient ewes were food restricted for 18 h before surgery. The fetus and umbilical cord were exposed and fetal blood was collected from the umbilical artery and vein, while maternal blood was collected from the uterine artery and vein ipsilateral to the fetus, with the resulting serum stored in −80 °C until further analysis. The fetus was then euthanized (88 mg/kg, Euthasol; VetOne), excised, and fetal weight, head circumference, crown-rump length, abdominal circumference, femur and tibia length were recorded. The fetal liver and pancreas were harvested, weighed, and snap frozen in liquid nitrogen. The ewe was euthanized (88 mg/kg, Euthasol; VetOne) and a complete hysterectomy was performed and all placentomes were excised and recorded for total placentome number and weight. Thirty placentomes were randomly selected and snap frozen in liquid nitrogen. The resulting tissue was pulverized using a mortar and pestle and stored at −80 °C for later use.

### 4.4. Biochemical Analysis of Blood Samples

Plasma glucose and lactate were measured by Yellow Spring Instrument 2900 (YSI Incorporated, Yellow Springs, OH), and plasma amino acids were measured by HPLC as described previously [14,15]. Maternal and fetal plasma concentrations of insulin and IGF1 were assessed by enzyme-linked immunosorbent assay (ALPCO Immunoassays, Salem, NH; 80-IN-SOV-E01 and 22-IGFHU-E01, respectively) as described previously [14,15]. The concentration of plasma CSH was assessed by radioimmunoassay (RIA) as described previously [13]. Maternal and fetal plasma glucagon concentrations were assessed by enzyme-linked immunosorbent assay (ALPCO; 48-GLUHU-E01), which was validated for use with sheep plasma and exhibited an intra-assay coefficient of variation ranging from 0.2 to 11.6%, for the highest to lowest plasma concentrations, respectively.

### 4.5. Cell Lines

Immortalized ovine trophoblast (iOTR) cells [16] were used to test the degree of RNAi of the SLC2A3 shRNA construct. To infect the cells, a frozen aliquot of SLC2A3 RNAi or NTS RNAi lentivirus was resuspended in 500 μL of DMEM-F12 medium [16] (supplemented with: 10% FBS, 1× penicillin-streptomycin-amphotericin B solution, 10 μg/mL insulin, 0.1 mM non-essential amino acids, 2 mM glutamine, and 1 mM sodium pyruvate) with 8 μg/mL polybrene (Sigma-Aldrich, St. Louis, MO, USA). The iOTR cells were incubated with lentiviral particles at a multiplicity of infection (MOI) of 500 for 8 h at 37 °C and 5% CO_2_, after which the transfection media was replaced with fresh complete media. The subsequent cells were passaged up to a 150-mm tissue culture plate, pelleted and stored in −80 °C until further analysis.

### 4.6. Western Blot Analysis

Cellular protein from 75 dGA placentomes was assessed using Western immunoblot analysis. Protein isolation and analysis were done in accordance with the methods described previously [14,15]. Pulverized placentome tissue (75 mg) was lysed in 500 μL of lysis buffer and sonicated on ice. For placental SLC2A3 analysis, 25 μg of protein from each sample were electrophoresed through NuPAGE 4–12% Bis Tris gels (Life Technologies, Carlsbad, CA, USA), and transferred to a 0.45 μm pore nitrocellulose membrane. For iOTR cell SLC2A3 analysis, 10 μg of protein from each sample were electrophoresed through NuPAGE 4–12% Bis-Tris gels (Life Technologies), and transferred to a 0.45 μm pore nitrocellulose membrane. The resulting blots were stained with Ponceau S (Sigma Aldrich, St. Louis, MO, USA) to assess total protein per lane using the ChemiDoc XRS+ (BioRad, Hercules, CA, USA). To visualize SLC2A3, the blots were incubated in a 1:1000 dilution of CSU-α-SCL2A3-22 [15] for 24 h at 4 °C. After washing, the blots were incubated in a 1:5000 dilution of goat α-rabbit IgG conjugated to horse radish peroxidase (ab97051; Abcam, Cambridge, MA, USA). Membranes were developed using an ECL Western Blotting Detection Reagent chemiluminescent kit (Amersham, Pittsburgh, PA, USA) and imaged using the ChemiDoc XRS+ (BioRad). Densitometry of SLC2A3 was normalized on the total protein per lane. To account for technical error between immunoblots, a common sample was included in each immunoblot and densitometry measurements were adjusted based on the average densitometry measurements of the common sample.

For analysis of placental SLC2A1, 5 μg of protein from each sample were electrophoresed through a 4–15% Tris-Glycine stain-free gel (BioRad) and transferred to a 0.45 μm pore nitrocellulose membrane. After transfer, the nitrocellulose membrane was imaged using the ChemiDoc XRS+ chemiluminescence system (BioRad) to assess the total protein per lane to use for normalization. To visualize SLC2A1, the blots were incubated in a 1:40,000 dilution of rabbit α-SLC2A1 (07-1401; EMD Millipore, Burlington, MA, USA) for 24 h at 4 °C. After washing, the blots were incubated in a 1:80,000 dilution of goat α-rabbit IgG conjugated to horse radish peroxidase (ab205718; Abcam). As described above, densitometry analysis of SLC2A1 was performed using Image Lab software (version 6.1; BioRad) and normalized on total protein/lane.

For analysis of placental SLC2A8, 20 μg of each sample were electrophoresed through 4–15% Tris-Glycine stain-free gels (BioRad) and transferred and analyzed as described for SLC2A1. SLC2A8 was visualized using a 1:2000 dilution of rabbit α-SLC2A8 (LS-C757596; LifeSpan BioSciences, Seattle, WA) and a 1:10,000 dilution of goat α-rabbit IgG conjugated to horse radish peroxidase (ab97051; Abcam). INSR was visualized using a 1:1000 dilution of mouse α-INSR-β (ab69058; Abcam) and a 1:5000 dilution of goat alpha-mouse IgG conjugated to horse radish peroxidase (ab6789; Abcam).

### 4.7. RNA Isolation

Total cellular RNA was isolated from 75 dGA pulverized placentome samples using the RNeasy Mini Kit (Qiagen, Hilden, Germany) according to the manufacturer’s protocol. RNA concentration was quantified using the BioTek Synergy 2 Microplate Reader (BioTek, Winooski, VT, USA), and RNA quality was measured by the 260/280 nm absorbance ratio. Samples were stored at −80 °C until use.

### 4.8. cDNA Synthesis and Quantitative Real-Time PCR

cDNA was generated from 2 μg of total cellular RNA using iScript Reverse Transcription Supermix (BioRad) according to the manufacturer’s protocol. To control for variance in the efficiency of the reverse transcription reaction, cDNA was quantified using the Quant-iT OliGreen ssDNA Assay Kit (Invitrogen, Carlsbad, CA, USA) according to the manufacturer’s protocol, and quality was measured by the 260/280 absorbance ratio. An equal mass of cDNA (10 ng/μL) was used for each sample in the quantitative real-time PCR (qRT-PCR) reaction. qRT-PCR was performed using the CFX384 Real-Time System (BioRad). Forward and reverse primers for qRT-PCR were designed using Oligo software (Molecular Biology Insights, Cascade, CO, USA) to amplify an intron-spanning product. Primer sequences and amplicon size are summarized in Table 7. Standard curves were generated as described previously [17]. Briefly, a PCR product for each gene was generated using cDNA from 135 dGA fetal placenta as a template and cloned into the StrataClone vector (Agilent Technologies), and each PCR product was sequenced to verify amplification of the correct cDNA. Using the PCR products amplified from the sequenced plasmids, standard curves were generated for each mRNA from 1 × 10^2^ to 1 × 10^−5^ pg, and were used to measure amplification efficiency. The starting quantity (pg) was normalized by dividing the starting quantity of mRNA of interest by the starting mRNA quantity (pg) of ribosomal protein S15 (RPS15) [17].

### 4.9. Statistical Analysis

Data were analyzed by two-way analysis of variance using GraphPad Prism (version 9) to analyze the main effects of treatment and fetal sex, as well as the treatment × sex interaction. The pregnancy success rate limited the final number, such that the study was not sufficiently powered to examine the effect of fetal sex. As there were no treatment by fetal sex interactions, the data are presented as the main effect of treatment only. Statistical significance was set at *p* ≤ 0.05 and a statistical tendency at *p* ≤ 0.10. Data are reported as the mean ± standard error of the mean (SEM).

## 5. Conclusions

Using lentiviral-mediated in vivo RNAi, we determined that a deficiency of SLC2A3, which is localized specifically to the microvillous apical membrane of placental trophoblasts in sheep, reduced the placental transfer of glucose to the fetus during the first-half of gestation resulting in impaired fetal growth and development. Furthermore, by impacting placental glucose uptake, placental function was altered in a fashion, which either directly or indirectly impacted maternal metabolic hormone secretion, highlighting the integration of the maternal, placental and fetal compartments of pregnancy.

## Figures and Tables

**Figure 1 ijms-23-12530-f001:**
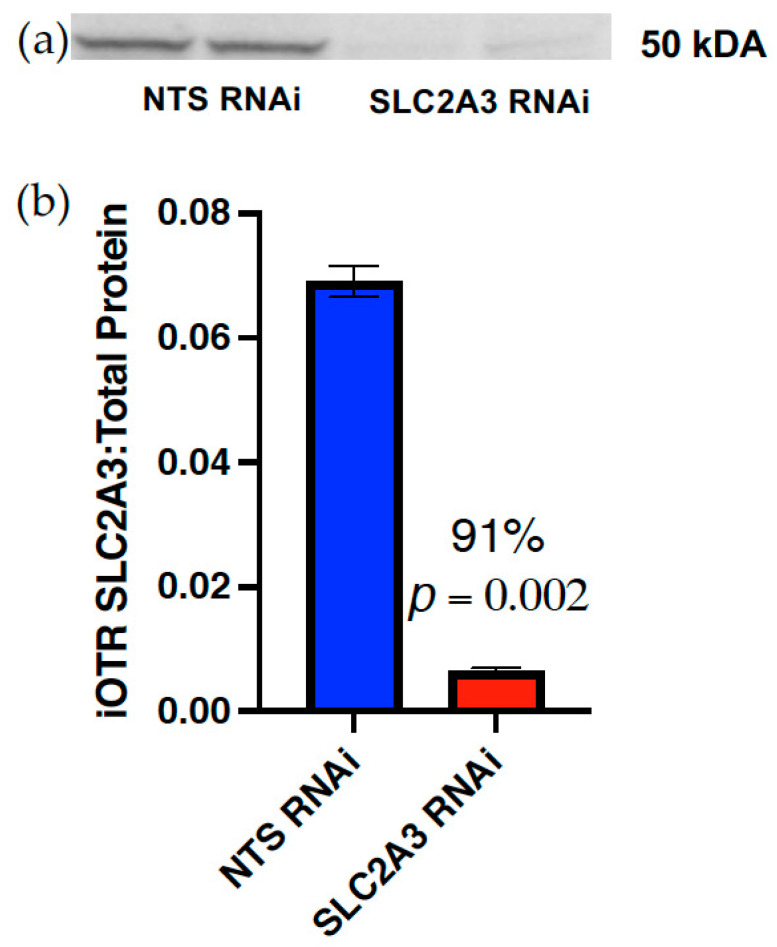
Efficiency of SLC2A3 RNAi in iOTR cells: (**a**) iOTR SLC2A3 detected by Western blot analysis following infection with either the NTS RNAi or SLC2A3 RNAi lentivirus; (**b**) concentration of SLC2A3, relative to total protein transferred, following infection with NTS RNAi or SLC2A3 RNAi lentivirus. Data are shown as means ± SEM. NTS, non-targeting sequence; RNAi, RNA interference.

**Figure 2 ijms-23-12530-f002:**
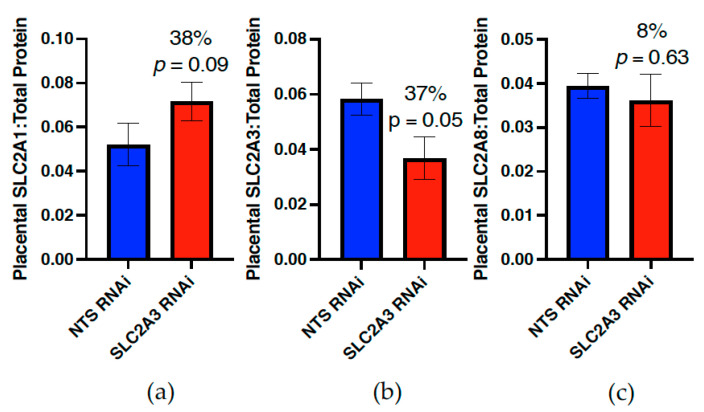
Impact of SLC2A3 RNAi on placental concentrations of (**a**) SLC2A1, (**b**) SLC2A3 and (**c**) SLC2A8 at mid-gestation in sheep. Data are shown as means ± SEM. NTS, non-targeting sequence; RNAi, RNA interference.

**Figure 3 ijms-23-12530-f003:**
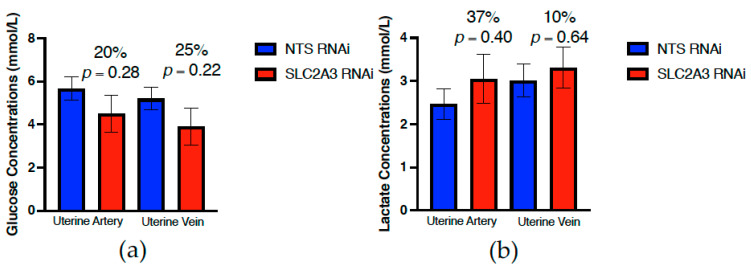
Impact of SLC2A3 RNAi on both uterine artery and vein concentrations of (**a**) glucose and (**b**) lactate in plasma samples harvested at 75 dGA. Data are shown as means ± SEM. NTS, non-targeting sequence; RNAi, RNA interference.

**Figure 4 ijms-23-12530-f004:**
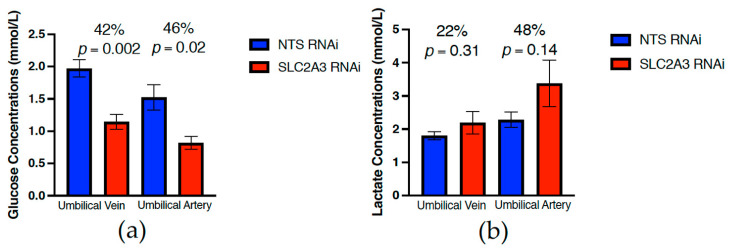
Impact of SLC2A3 RNAi on both umbilical vein and artery concentrations of (**a**) glucose and (**b**) lactate in plasma samples harvested at 75 dGA. Data are shown as means ± SEM. NTS, non-targeting sequence; RNAi, RNA interference.

**Figure 5 ijms-23-12530-f005:**
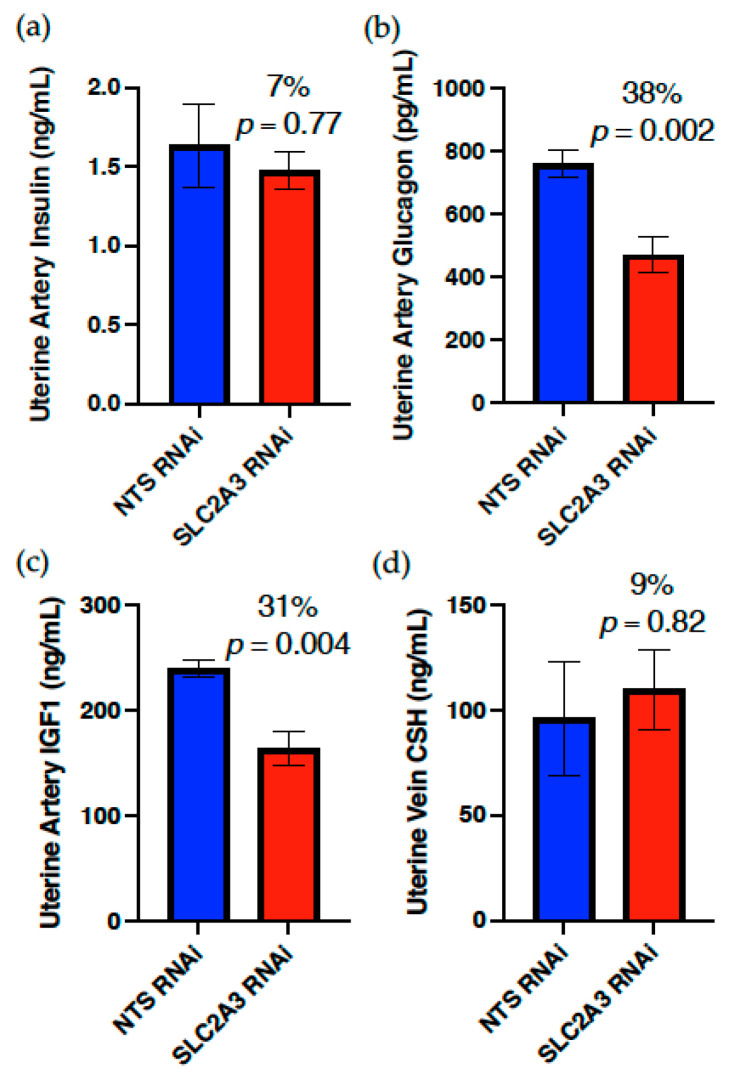
Impact of SLC2A3 RNAi on uterine artery concentrations of (**a**) insulin, (**b**) glucagon and (**c**) IGF1, and (**d**) uterine vein concentrations of CSH. Data are shown as means ± SEM. NTS, non-targeting sequence; RNAi, RNA interference.

**Figure 6 ijms-23-12530-f006:**
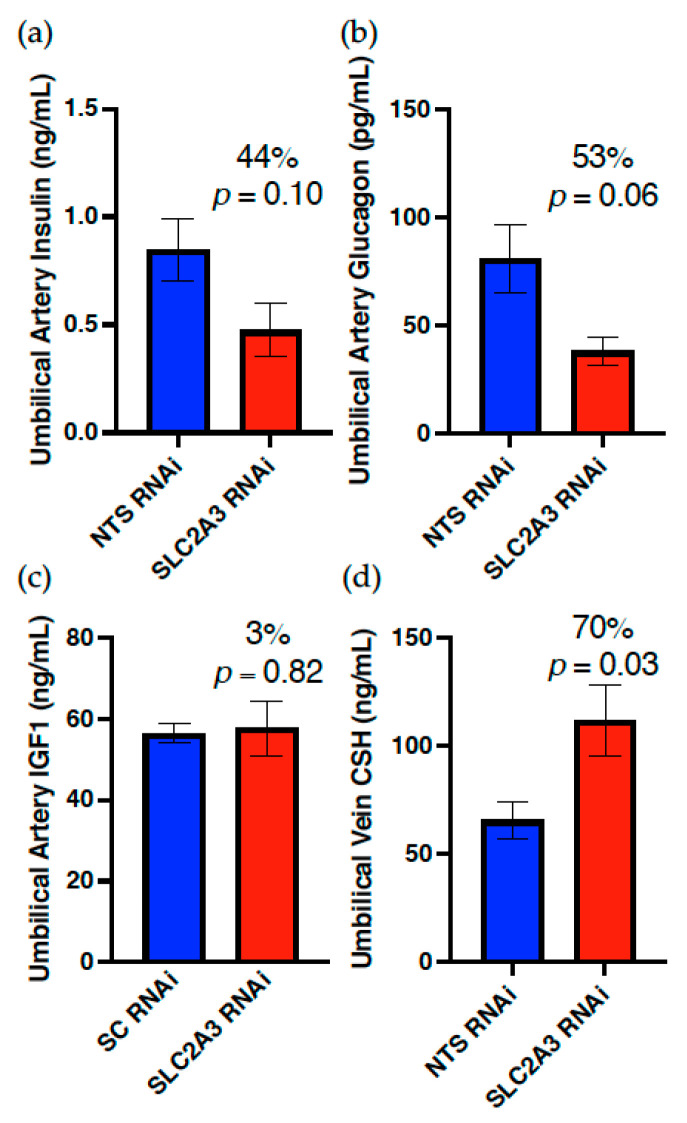
Impact of SLC2A3 RNAi on umbilical artery concentrations of (**a**) insulin, (**b**) glucagon, and (**c**) IGF1, and (**d**) umbilical vein concentrations of CSH. Data are shown as means ± SEM. NTS, non-targeting sequence; RNAi, RNA interference.

**Figure 7 ijms-23-12530-f007:**
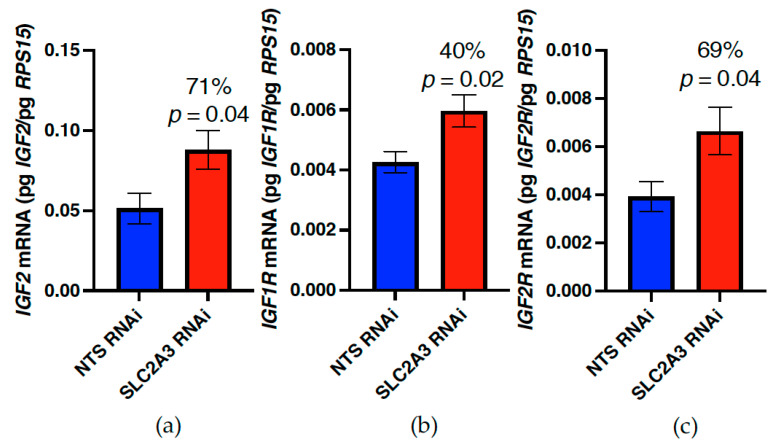
Impact of SLC2A3 RNAi on placental concentrations of (**a**) IGF2, (**b**) IGF1R, and (**c**) IGF2R mRNA. Data are shown as means ± SEM. NTS, non-targeting sequence; RNAi, RNA interference.

**Table 1 ijms-23-12530-t001:** Fetal and Doppler velocimetry measurements assessed at 70 dGA.

	NTS RNAi	SLC2A3 RNAi	*p*-Value	% Change
Binocular Distance, cm	3.53 ± 0.10	3.29 ± 0.06	0.06	6.68
Crown-rump length, cm	14.88 ± 1.13	14.31 ± 0.40	0.58	3.83
Abdominal circumference, cm	13.09 ± 0.55	12.20 ± 0.46	0.24	6.82
Femur Length, cm	2.97 ± 0.12	2.34 ± 0.18	0.01	21.19
Tibia Length, cm	2.70 ± 0.20	1.99 ± 0.18	0.03	26.38
Pulsatility Index	2.80 ± 0.27	2.97 ± 0.15	0.59	6.15
Resistance Index	0.86 ± 0.05	0.85 ± 0.04	0.92	0.79
Systolic: Diastolic	16.18 ± 6.60	11.04 ± 3.96	0.52	31.76
Fetal heart rate, bpm	204.72 ± 1.91	200.03 ± 8.69	0.61	2.29
Umbilical artery cross-sectional area, cm^2^	0.09 ± 0.007	0.104 ± 0.01	0.44	11.52
Umbilical artery cross-sectional diameter, cm	0.34 ± 0.01	0.35 ± 0.02	0.74	2.37

Data are shown as means ± SEM for all ewes in each treatment group. NTS, non-targeting sequence; RNAi, RNA interference.

**Table 2 ijms-23-12530-t002:** Placental and fetal measurements obtained at 75 dGA.

	NTS RNAi	SLC2A3 RNAi	*p*-Value	% Change
Fetal weight, g	208.61 ± 9.93	179.89 ± 10.84	0.08	13.77
Head circumference, cm	13.90 ± 0.19	12.88 ± 0.43	0.05	7.37
Crown-rump length, cm	19.33 ± 0.4	19.17 ± 0.36	0.76	0.86
Abdominal circumference, cm	13.62 ± 0.51	12.92 ± 0.35	0.28	5.14
Femur length, cm	4.33 ± 0.20	3.67 ± 0.17	0.03	15.22
Tibia length, cm	3.42 ± 0.20	2.78 ± 0.19	0.05	18.54
Liver weight, g	13.21 ± 0.95	12.65 ± 1.06	0.71	4.20
Pancreas weight, mg	470.00 ± 34.35	363.33 ± 15.85	0.02	22.70
Placentome weight, g	536.10 ± 54.05	423.39 ± 34.81	0.13	21.02
Placentome number	81.83 ± 5.51	73.33 ± 7.17	0.37	10.39

Data are shown as mean values ± SEM for all ewes in each treatment group. NTS, non-targeting sequence; RNAi, RNA interference.

**Table 3 ijms-23-12530-t003:** Maternal plasma amino acid concentrations (75 dGA).

	NTS RNAi Uterine Artery	SLC2A3 RNAi Uterine Artery	*p*-Value	% Change	NTS RNAi Uterine Vein	SLC2A3 RNAi Uterine Vein	*p*-Value	% Change
TAU	43.50 ± 8.70	49.55 ± 8.83	0.64	13.91	78.85 ± 17.57	74.16 ± 17.13	0.98	0.92
ASP	18.53 ± 2.66	17.65 ± 2.58	0.82	4.78	27.42 ± 3.50	34.10 ± 10.13	0.55	24.38
THR	108.92 ± 15.67	147.58 ± 17.81	0.13	35.50	153.15 ± 45.77	151.47 ± 19.38	0.97	1.10
SER	63.61 ± 6.28	75.04 ± 7.41	0.27	17.97	109.98 ± 50.80	71.67 ± 8.15	0.47	34.84
ASN	26.70 ± 3.15	42.09 ± 3.93	0.01	57.67	32.27 ± 5.42	42.23 ± 5.13	0.21	30.85
GLU	105.34 ± 11.72	106.04 ± 6.40	0.96	0.66	158.64 ± 12.88	172.02 ± 34.99	0.73	8.43
GLN	309.41 ± 17.70	317.86 ± 25.19	0.79	2.73	308.16 ± 30.39	328.82 ± 28.26	0.63	6.70
PRO	78.25 ± 9.15	81.99 ± 5.96	0.74	4.78	84.70 ± 6.03	89.16 ± 6.52	0.63	5.27
GLY	628.54 ± 53.55	578.48 ± 32.09	0.44	7.96	630.56 ± 63.84	687.23 ± 66.76	0.55	8.99
ALA	198.69 ± 14.19	219.90 ± 10.41	0.26	10.68	219.66 ± 19.76	245.43 ± 13.65	0.31	11.73
CIT	188.30 ± 23.08	256.7 ± 34.01	0.13	36.32	171.15 ± 19.28	253.27 ± 35.50	0.07	47.98
VAL	197.76 ± 13.18	231.82 ± 13.41	0.10	17.22	201.25 ± 24.21	219.92 ± 12.70	0.51	9.28
CYS	21.92 ± 2.88	17.99 ± 3.70	0.42	17.93	11.94 ± 4.15	21.28 ± 4.84	0.17	78.20
MET	22.15 ± 1.96	27.58 ± 2.98	0.16	24.51	33.38 ± 10.20	28.49 ± 3.39	0.66	14.65
ILE	102.26 ± 3.79	115.23 ± 6.93	0.13	12.69	94.24 ± 4.77	104.06 ± 3.18	0.12	10.42
LEU	120.45 ± 4.05	134.11 ± 9.62	0.22	11.33	114.62 ± 9.81	125.2 ± 6.66	0.39	9.23
TYR	39.20 ± 3.28	49.47 ± 5.61	0.14	26.22	52.23 ± 12.11	54.75 ± 7.68	0.86	4.82
PHE	34.06 ± 1.89	38.38 ± 3.34	0.29	12.66	44.80 ± 10.34	42.56 ± 4.15	0.84	4.99
TRP	34.27 ± 1.78	36.78 ± 2.54	0.44	7.30	35.72 ± 4.65	39.95 ± 1.99	0.96	0.65
ORN	94.49 ± 6.14	115.83 ± 8.20	0.06	22.58	97.07 ± 17.58	105.10 ± 8.02	0.69	8.27
LYS	103.41 ± 6.97	144.09 ± 17.23	0.05	39.33	117.46 ± 15.91	148.69 ± 19.41	0.24	26.59
HIS	59.89 ± 1.70	61.65 ± 3.16	0.63	2.94	57.60 ± 2.77	62.99 ± 3.74	0.27	9.38
ARG	210.83± 9.11	233.19 ± 20.86	0.35	10.61	219.04 ± 11.37	231.31 ± 23.04	0.64	5.60

Data are shown as mean values ± SEM for all ewes in each treatment group. TAU, taurine; ASP, aspartic acid; THR, threonine; SER, serine; ASN, asparagine; GLU, glutamic acid; GLN, glutamine; PRO, proline; GLY, glycine; ALA, alanine; CIT, citrulline; VAL, valine; CYS, cysteine; MET, methionine; ILE, isoleucine; LEU, leucine; TYR, tyrosine; PHE, phenylalanine; TRP, tryptophan; ORN, ornithine; LYS, lysine, HIS, histidine; ARG, arginine.

**Table 4 ijms-23-12530-t004:** Fetal plasma amino acid concentrations (75 dGA).

	NTS RNAi Umbilical Art.	SLC2A3 RNAi Umbilical Art.	*p*-Value	% Change	NTS RNAi Umbilical Vein	SLC2A3 RNAi Umbilical Vein	*p*-Value	% Change
TAU	132.41 ± 16.08	141.54 ± 19.40	0.73	6.89	132.14 ± 13.98	144.70 ± 23.33	0.64	9.50
ASP	43.10 ± 4.27	41.54 ± 3.05	0.80	3.63	40.87 ± 3.80	32.67 ± 2.58	0.12	20.06
THR	453.57 ± 32.87	561.20 ± 66.67	0.15	23.73	497.44 ± 24.57	531.22 ± 57.58	0.58	6.79
SER	375.99 ± 25.84	420.26 ± 48.66	0.40	11.77	328.51 ± 15.49	338.54 ± 38.41	0.80	3.05
ASN	52.01 ± 2.62	66.88 ± 5.40	0.02	28.60	76.66 ± 2.62	86.04 ± 6.95	0.21	12.24
GLU	182.77 ± 23.49	181.57 ± 15.68	0.97	0.65	49.03 ± 3.88	40.59 ± 8.67	0.37	17.22
GLN	453.66 ± 36.48	495.05 ± 38.65	0.47	9.12	591.28 ± 28.78	589.07 ± 32.74	0.96	0.37
PRO	114.12 ± 14.32	136.71 ± 9.92	0.28	19.80	143.05 ± 12.36	163.25 ± 9.12	0.24	14.12
GLY	465.10 ± 46.10	458.02 ± 48.14	0.92	1.52	550.47 ± 30.18	477.85 ± 26.96	0.11	13.19
ALA	319.75 ± 28.75	349.43 ± 50.04	0.59	9.28	417.10 ± 18.71	384.69 ± 17.40	0.24	7.77
CIT	151.75 ± 16.05	184.56 ± 19.39	0.23	21.62	152.28 ± 16.16	168.38 ± 19.32	0.54	10.57
VAL	263.06 ± 19.26	313.77 ± 42.27	0.25	19.28	318.44 ± 26.91	332.76 ± 36.17	0.75	4.50
CYS	16.52 ± 1.99	19.53 ± 2.05	0.34	18.25	12.72 ± 2.86	12.39 ± 1.50	0.93	2.59
MET	79.89 ± 6.43	89.07 ± 11.02	0.46	11.48	98.43 ± 6.07	103.53 ± 4.21	0.53	5.18
ILE	75.66 ± 4.25	91.57 ± 11.79	0.18	21.03	106.30 ± 5.92	107.65 ± 10.54	0.91	1.27
LEU	130.62 ± 7.42	150.52 ± 17.26	0.26	15.24	188.21 ± 9.92	181.61 ± 14.04	0.70	3.51
TYR	104.88 ± 3.21	116.85 ± 14.31	0.35	11.42	134.05 ± 9.35	136.81 ± 14.15	0.87	2.05
PHE	92.42 ± 2.39	108.34 ± 11.15	0.13	17.22	123.35 ± 5.79	127.38 ± 7.21	0.67	3.27
TRP	50.73 ± 1.83	45.51 ± 4.21	0.23	10.28	59.73 ± 2.75	55.44 ± 2.72	0.30	7.18
ORN	160.01 ± 15.14	196.74 ± 43.89	0.38	22.95	165.92 ± 11.63	186.65 ± 39.12	0.59	12.49
LYS	202.84 ± 16.16	210.34 ± 19.88	0.78	3.70	260.43 ± 15.82	270.66 ± 30.56	0.76	3.93
HIS	50.08 ± 3.82	52.69 ± 4.12	0.66	5.20	68.09 ± 1.68	66.35 ± 7.37	0.81	2.55
ARG	251.54 ± 15.84	193.81 ± 9.20	0.03	22.95	313.01 ± 19.29	246.22 ± 15.08	0.03	21.34

Data are shown as mean values ± SEM for all ewes in each treatment group. TAU, taurine; ASP, aspartic acid; THR, threonine; SER, serine; ASN, asparagine; GLU, glutamic acid; GLN, glutamine; PRO, proline; GLY, glycine; ALA, alanine; CIT, citrulline; VAL, valine; CYS, cysteine; MET, methionine; ILE, isoleucine; LEU, leucine; TYR, tyrosine; PHE, phenylalanine; TRP, tryptophan; ORN, ornithine; LYS, lysine, HIS, histidine; ARG, arginine.

**Table 5 ijms-23-12530-t005:** Placental insulin-like growth factor mRNA concentrations (75 dGA).

mRNA	NTS RNAi	SLC2A3 RNAi	*p*-Value	% Change
*IGF1,* pg/pg	0.0011 ± 0.0002	0.0011 ± 0.00013	0.88	3.18
*IGFBP1*, pg/pg	0.00018 ± 0.000076	0.00013 ± 0.000066	0.64	26.81
*IGFBP2*, pg/pg	0.00051 ± 0.000082	0.00057 ± 0.000057	0.57	11.58
*IGFBP3*, pg/pg	0.025 ± 0.0056	0.028 ± 0.0049	0.69	12.19

Data are shown as mean values ± SEM for the starting quantity of the mRNA of interest (pg) divided by the starting quantity (pg) of the housekeeping mRNA (*RPS15*).

**Table 6 ijms-23-12530-t006:** Non-targeting sequence (NTS) RNAi and SLC2A3 RNAi shRNA sequences.

Oligonucleotide	Sequence (5′–3′)
NTS shRNA sense strand	GAGTTAAAGGTTCGGCACGAATTCAAGAGATTCGTGCCGAACCTTTAACTC
SLC2A3 shRNA sense strand	GCGCAACTCAATGCTTATTGTTTCAAGAGAACAATAAGCATTGAGTTGCGC

**Table 7 ijms-23-12530-t007:** Primers and product sizes for cDNA used in qRT-PCR.

cDNA	Forward Primer (5′–3′)	Reverse Primer (5′–3′)	Product, bp
*RPS15*	ATCATTCTGCCCGAGATGGTG	TGCTTGACGGGCTTGTAGGTG	134
*IGF1*	TCGCATCTCTTCTATCTGGCCCT	ACAGTACATCTCCAGCCTCCTCA	240
*IGF2*	GACCGCGGCTTCTACTTCAG	AAGAACTTGCCCACGGGGTAT	203
*IGFBP1*	TGATGACCGACTCCAGTGAG	GTCCAGCGAAGTCTCACAC	248
*IGFBP2*	CAATGGCGAGGAGCACTCTG	TGGGGATGTGTAGGGAATAG	330
*IGFBP3*	CTCAGACGACAGACACCCA	GGCATATTTGAGCTCCAC	336
*IGF1R*	AACTGTCATCTCCAACCTC	CAAGCCTCCCACTATCAAC	493
*IGF2R*	GACTTGTGTCCAGACCAGATTC	GCCGTCGTCCTCACTCTCATC	674

## Data Availability

Data available upon request to the corresponding author.

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
