# Peer review of "Impact of Placental SLC2A3 Deficiency during the First-Half of Gestation"

_ijms, 2022, doi:10.3390/ijms232012530_

Round 1

Reviewer 1 Report

The manuscript entitled “Impact of placental SLC2A3 deficiency during the first half of Gestation “authored by Lynch et al., is a very interesting work.

Authors suggested the deficit of SLC2A3 would have a major impact on placental uptake of glucose, placental development and function, and fetal development.

Using Lentiviral-mediated RNAi to attenuate the expression of placental SLC2A3, they assessed the relative importance in placental glucose transport during the gestation period in sheep as a mammalian model.

I recommended major revision

Accordingly, we hypothesized that SLC2A3 deficiency would result in impaired placental development and significant FGR by mid-gestation

Line 59, add full stop “….in sheep. The sheep ………..

I think, the authors should revise all the probability such (P<0.10) should change to (P<0.01).

The abbreviations are above the tables should be capital letter like the table.

Why the tables seem pictures?

It’s not impotent to rewrite the figures names in the discussions

Author Response

We thank this referee for his/her positive comments about our manuscript, and have addressed each comment separately below. 

The manuscript entitled “Impact of placental SLC2A3 deficiency during the first half of Gestation “authored by Lynch et al., is a very interesting work.

Authors suggested the deficit of SLC2A3 would have a major impact on placental uptake of glucose, placental development and function, and fetal development.

Using Lentiviral-mediated RNAi to attenuate the expression of placental SLC2A3, they assessed the relative importance in placental glucose transport during the gestation period in sheep as a mammalian model.

I recommended major revision

Accordingly, we hypothesized that SLC2A3 deficiency would result in impaired placental development and significant FGR by mid-gestation

Line 59, add full stop “….in sheep. The sheep ………..

We agree that as initially worded, there was redundancy within the sentence.  As such, we deleted “in sheep”, which makes this compound sentence now read as intended.

I think, the authors should revise all the probability such (P<0.10) should change to (P<0.01).

That would not be correct or honest.  As stated in line 467 “Statistical significance was set at P£0.05 and a statistical tendency at P£0.10.”  The use of P£0.05 as the cutoff for statistically significant differences and P£0.10 as the cutoff for statistical tendencies, is a widely accepted approach in the biomedical literature.  Keeping in mind, actual “statistical significance” is an individual judgement call, and is not actually defined or set by statistical principles. This is why we used the actual P values for individual comparisons within the figures and tables, so the reader could make that judgement call on whether the difference was “meaningful.”

The abbreviations are above the tables should be capital letter like the table.

We have to state that we are a bit confused by this comment, as in reviewing the submitted manuscript, any abbreviations in the table titles were capitalized the same way as the abbreviations within the table.  However, we did recognize that in the subscript for Tables 3 and 4, that the individual amino acid abbreviations were not capitalized as they were in the table.  This has been corrected.

Furthermore, during this process, we recognized that there was an issue with the two tables associated with the Materials and Methods section (Table 6 was misnumbered, and the wrong table was inserted for Table 7), which have now been corrected.  So we thank this referee for drawing our attention to our tables.

Why the tables seem pictures?

Tables 3 and 4 were inserted into the manuscript in this fashion in order to make these large tables fit.  The original tables were created in the “landscape orientation”, which we were unabled to insert them into this manuscript directly in that fashion.  We are happy to work with the editor to fix this, if their current appearance is unacceptable.

It’s not impotent to rewrite the figures names in the discussions

While we understand this comment from the referee, we feel that by identifying the Figure we are discussing within the Discussion section, helps draw the reader’s attention to the correct figure, thereby making the Discussion easier to read.  Again, we will remove these if the editor requests it.

Reviewer 2 Report

This paper investigated the SLC2A3 deficiency effect on the first half of gestation in this paper. T e authors found the deficiency of SLC2A3 resulted in fetal hypoglycemia, reduced fetal development, and altered metabolic hormone concentrations. Overall, it is an interesting paper and provides some of the mechanisms. I recommend it be published if the authors address the main points pointed out here. T e followings are the comments: 

1. The detailed mechanism of the SLC2A3 deficiency is unclear in this paper. The test of placental mRNA concentration of the insulin-like growth factor axis is not enough to show the mechanism. The authors should do more experiments to clarify how SLC2A3 impacts gestation. 

2. In vitro studies are essential to introduce the mechanism of the SLC2A3 deficiency effect. T e authors should use some cell lines to confirm the mechanism if possible. 

3.   ease provide a schematic diagram to show the potential mechanism of the SLC2A3 deficiency.

Author Response

This paper investigated the SLC2A3 deficiency effect on the first half of gestation in this paper. T e authors found the deficiency of SLC2A3 resulted in fetal hypoglycemia, reduced fetal development, and altered metabolic hormone concentrations. Overall, it is an interesting paper and provides some of the mechanisms. I recommend it be published if the authors address the main points pointed out here. T e followings are the comments: 

We thank this referee for his/her positive comments about our manuscript, and have addressed each comment separately below.  However, we take issue with the assessment that the manuscript requires “Extensive editing of English language and style required.”  All of the authors, except Dr. Ali, are native English speakers, educated in U.S. institutions.  The manuscript was carefully edited for proper English usage, spelling and grammar before submission, so unless there are specific issues identified, we cannot address this.

  1. The detailed mechanism of the SLC2A3 deficiency is unclear in this paper. The test of placental mRNA concentration of the insulin-like growth factor axis is not enough to show the mechanism. The authors should do more experiments to clarify how SLC2A3 impacts gestation.

SLC2A3 (GLUT3) is a facilitative glucose transporter, and in this case is located on the apical trophoblast membrane of the sheep placenta.  The mechanism by which SLC2A3 transports glucose into a cell has been examined some time ago, as with most of the facilitative glucose transporters.  The mechanism impacted in these studies is glucose uptake by the placenta, thereby limiting the availability of glucose to be transported to the fetus.  As 80% of the glucose taken up by the placenta during the first half of gestion is oxidized by the placenta, reducing glucose uptake by creating an SLC2A3 deficiency not only impacts fetal availability of glucose, but also glucose available for placental oxidative metabolism, thereby impacting its function.  The examination of IGF axis mRNA can simply be viewed as a “read out” of altered placental function in response to impaired glucose uptake.  We believe we have more than adequately addressed these points in the manuscript already.

Doing more experiments to “clarify how SLC2A3 impacts gestation” is something that we would like to do.  However, we hope that this referee recognizes that conducting this type of in vivo experiments are very costly, time consuming and labor intensive.  Additional funding will be required to generate adequate SLC2A3 RNAi and NTS RNAi pregnancies to allow them to gestate to near-term, when we can catheterize both maternal and fetal vasculature, allowing the steady-state assessment of maternal and fetal physiology.  Our current findings provide a lot of new insight into the necessity of placental SLC2A3, and sets the stage for doing the more intensive in vivo studies.

  1. In vitro studies are essential to introduce the mechanism of the SLC2A3 deficiency effect. T e authors should use some cell lines to confirm the mechanism if possible. 

There has already been a lot of in vitro studies examining the mechanisms of various facilitative glucose transporters, such as SLC2A3.  We do plan to use both overexpression and RNAi of SLC2A3 and SLC2A1 in our immortalized oTR cells, and then subject these to O2K respirometry, but obviously this cannot be accomplished in the 10-day time frame allotted for revising this manuscript.  Furthermore, such in vitro experiments would not provide a clear delineation of the impact of SLC2A3 deficiency in the placenta, since the trophoblast cells that could be used (either primary or immortalized lines) are not polarized like the placental epithelium, so you would not have the distinct localization of SLC2A3 and SLC2A1.  While in vitro mechanistic studies are important, as pointed out in the Discussion, pregnancy is a three-compartment system (maternal, placental and fetal) and all three have to be studied to understand the impact of altering something within even 1 of the 3 compartments.  That is why the in vivo studies like we report are so critical to our understanding of pregnancy physiology. 

  1. ease provide a schematic diagram to show the potential mechanism of the SLC2A3 deficiency.

We provided a graphical abstract with our submission, which we believe is adequate, and highlights the major findings.  Again, the mechanism of SLC2A3 deficiency is to inhibit placental uptake of glucose.

Round 2

Reviewer 1 Report

authors addressed all the comments. So, no further comments are required. The manuscript can be accepted in the present form.